# Algebraic Positional Encodings

**Konstantinos Kogkalidis**[1,2]
kokos.kogkalidis@aalto.fi

**Jean-Philippe Bernardy**[3,4]
jean-philippe.bernardy@gu.se

**Vikas Garg**[1,5]
vgarg@csail.mit.edu

[1]Aalto University
[2]University of Bologna
[3]University of Gothenburg
[4]Chalmers University of Technology
[5]YaiYai Ltd

## Abstract

We introduce a novel positional encoding strategy for Transformer-style models, addressing the shortcomings of existing, often *ad hoc*, approaches. Our framework implements a flexible mapping from the algebraic specification of a domain to a positional encoding scheme, where positions are interpreted as orthogonal operators. This design preserves the structural properties of the source domain, thereby ensuring that the end-model upholds them. The framework can accommodate various structures, including sequences, grids and trees, but also their compositions. We conduct a series of experiments demonstrating the practical applicability of our method. Our results suggest performance on par with or surpassing the current state of the art, without hyper-parameter optimizations or "task search" of any kind. Code is available through https://aalto-quml.github.io/ape/.

## 1 Introduction

Attention-based models inheriting from the Transformer [Vaswani et al., 2017] have become ubiquitous in neural computation, supplanting the go-to models of the last decade and driving a continuous stream of breakthroughs across diverse domains. Their success is perhaps at odds with the Transformer's structural lenience. Its key building block, dot-product attention, is by default unable to perceive and utilize the structure and arrangement of the input/output tokens being processed. To address this limitation, a plethora of works have sought to endow Transformers with appropriate inductive biases. The default strategy is to adjust token representations via so-called *positional encodings*; vector operations that hint at the structure being modeled. Nonetheless, most positional encoding schemes to date are either empirically motivated, or tailored to specific tasks. This renders their theoretical evaluation challenging, and hinders any prospects of a unifying framework.

In this study, we seek to fill this gap with a theory-first approach. Through the lens of group theory, we scrutinize some of the most commonly targeted data structures, and express them by means of inductive definitions that reveal and explicate their structural properties. Leveraging this analysis, our modeling strategy invokes a homomorphic interpretation that maps each domain into **algebraic positional encodings** (APE): attention-compatible vector operations parameterizing (subgroups of) the orthogonal group. In the sequential context, algebraic positional encodings streamline the widely adopted rotary encodings of Su et al. [2023], while also offering clear theoretical insights on their success. More importantly, algebraic positional encodings naturally extend to non-sequential domains, such as $\kappa$-ary trees and multidimensional regular grids, paving the way for a simple and elegant methodology for interpretable and domain-general structurally-refined Transformers. We carry out an experimental evaluation in settings that allow for reproducible and statistically sound conclusions. Across the tasks considered, algebraic positional encodings consistently and significantly outperform strong baselines at an aggregate level, providing initial but compelling evidence that they constitute not just a *sensible meta-theory* for positional encodings, but also an *actionable alternative* to the current state of the art.

38th Conference on Neural Information Processing Systems (NeurIPS 2024).

§2 BACKGROUND
We introduce the problem (§2.1) and the vocabulary of the solution (§2.2).

§3 THEORY
We provide an algebraic characterization of positions in the context of different ambient structures. We frame algebraic positional encodings as structure-preserving semantic interpretations, and present reference implementations. Concretely:
  §3.1 *Sequences* are an isomorphism of the free group $\langle \mathbb{1} \rangle$ (*i.e.*, the integers, $\mathbb{Z}$), and can be interpreted as a single generator subgroup of the orthogonal group $O(d)$.
  §3.2 *Rotary Positional Encodings* correspond to a (quite literally) special case of this interpretation: $SO(d)$.
  §3.3 *k-ary Trees* are an isomorphism of the finitely generated group $\langle \mathbb{1}, \mathbb{2} \dots \kappa \rangle$, and can be interpreted as a finitely generated subgroup of $O(d)$.
  §3.4 *Regular Grids* are the group direct sum of multiple sequences. They can be interpreted as the matrix direct sum of their components' interpretations.
  §3.5 *Extensions* can be obtained in multiple directions.

§4 PRACTICE
We carry out fair and replicable experiments across all three structures analyzed, namely *Sequence Transduction* (§4.1), *Tree Transduction* (§4.2) and *Image Recognition* (§4.3), and find that algebraic positional encodings consistently match or outperform alternative schemes in the tasks considered (§4.4).

§6 LIMITATIONS
We close with a brief discussion of possible limitations and potential future improvements.

Table 1: A summary of this paper.

## 2 Background

### 2.1 The Problem with Dot-Product Attention

All transformer variants employ some variation of the multi-head scaled dot-product attention mechanism of Vaswani et al. [2017]. For each attention head, the dot-product attention between queries $\boldsymbol{X} \in \mathbb{R}^{m \times d}$ and keys $\boldsymbol{Y} \in \mathbb{R}^{n \times d}$ is defined as:

$$\text{atn}(\boldsymbol{X}, \boldsymbol{Y}) := \text{softmax}_{(n)} \left( \frac{(\boldsymbol{X}\boldsymbol{\Phi}^{(q)})(\boldsymbol{Y}\boldsymbol{\Phi}^{(k)})^\top}{\sqrt{d}} \right) \boldsymbol{Y}\boldsymbol{\Phi}^{(v)} \tag{1}$$

In equation (1), matrices $\boldsymbol{\Phi}^{(q)}, \boldsymbol{\Phi}^{(k)}, \boldsymbol{\Phi}^{(v)} : \mathbb{R}^{d \times d}$ enact linear functions, applied point-wise (broadcasted) across all $m$ and $n$ entries of $\boldsymbol{X}$ and $\boldsymbol{Y}$. The dot-product term $(\boldsymbol{X}\boldsymbol{\Phi}^{(q)})(\boldsymbol{Y}\boldsymbol{\Phi}^{(k)})^\top$ contains unnormalized attention scores in the Cartesian product of queries and keys. Unmodified, dot-product attention is permutation *invariant* with respect to its second argument; that is, for any arbitrary permutation $p_n \in \mathcal{S}_n$:

$$\text{atn}(\boldsymbol{X}, \boldsymbol{Y}) \equiv \text{atn}(\boldsymbol{X}, p_n(\boldsymbol{Y})) \tag{2}$$

Unless one is dealing with orderless structures like multisets or fully connected graphs, this property is generally undesirable. The lack of structural biases is traditionally counteracted by the component-wise addition of unidimensional periodic signals of varying frequencies. These, however, often prove inadequate in data-scarce domains, where extensive pretraining is impossible, and structure-rich domains, where a sequence-of-tokens projection is too radical of a simplification.

### 2.2 Recap on Group Theory

To address this issue, we propose an algebraic treatment of positional encodings, based on principles lent from group theory. For the sake of convenience and accessibility, we provide a brief recap of the notions of interest here. A *group* $G$ consists of a set of *elements* and a *binary operation* (\_·\_) satisfying four fundamental laws:

- The group is *closed* under the the group operation. For all $a$, $b$ in $G$, $a \cdot b$ is also in $G$.
- The group operation is *associative*. For all $a$, $b$, $c$ in $G$, $(a \cdot b) \cdot c = a \cdot (b \cdot c)$.
- The group operation has an *identity* element $e$, such that for all $a$ in $G$, $a \cdot e = e \cdot a = a$.
- Each group member has an *inverse*. For all $a$ in $G$, there exists some element $\bar{a}$ such that $a\bar{a} = \bar{a}a = e$, where $e$ is the identity element.

A group is characterized as *finite* or *infinite* depending on the number of elements it has. If all elements of a group $G$ can be expressed as a combination of a subset $S$ of the group elements (combined by means of the group operation, applied either on the elements themselves or on their inverses), we write $G = \langle S \rangle$. We say that $G$ is *generated* by $S$, and we call the elements of $S$ the *generators* of $G$. A group with a single generator is called *cyclic*.

## 3 The Algebra(s) of Positions

Our objective is to establish a framework that offers general and extensible *semantics* for positions across various structures – what we commonly encounter in the literature as *positional encodings*. Most existing proposals adopt a rather parochial stance, relying on maneuvers or heuristics tailored to specific applications and driven, predominantly, by extensive empirical investigations. As such, they fall short with respect to accommodating or reflecting the properties of the underlying structure. We follow a different approach. We adopt Montague's perspective, succinctly paraphrased as:

> "*syntax is an algebra, semantics is an algebra, and meaning is a homomorphism between them*" [Janssen, 2014].

We begin by noting that "positions" do not exist in isolation, but only in the context of some underlying ambient structure. We contend that reasonable positional encodings (*semantics*) may only be reliably obtained by taking into account exactly this structure, its formation rules and properties (*syntax*), and then applying an appropriate interpretation (*meaning*). This is *not* just an academic exercise: a careful syntactic specification is a prerequisite if we aim for semantics that adhere to certain properties, which is arguably preferable to searching for these properties in the wild.

### 3.1 Sequences

**Syntax**  We start from the simplest structure, and incidentally also the most standard one: the sequence. The full range of positions a token can occupy within a sequence coincides exactly with the naturals, $\mathbb{N}$. Relative paths $\mathbb{P}$ between any two positions can then be seen as the integers, $\mathbb{Z}$, with positive (resp. negative) numbers denoting forward (resp. backward) offsets. Using this insight, it is handy to inspect how the standard inductive definition of the integers provides the building blocks for path formation. We start with two constants: the empty path ($\mathbb{0}$), which relates any given point to itself, and the unit path ($\mathbb{1}$), which relates any point to its immediate next. We may compose simple paths into complex ones with the aid of a binary operation $+_{\mathbb{P}}$. This already suffices to specify all forward offsets. In order to construct backward offsets, we need a unary operation $(-)_{\mathbb{P}}$, such that $-\rho$ denotes the inverse of $\rho$. We can summarize the above by the grammar:

$$\mathbb{P} := \mathbb{0} \mid \mathbb{1} \mid \mathbb{P} +_{\mathbb{P}} \mathbb{P} \mid -\mathbb{P} \tag{3}$$

For this to make sense, the operations must be *coherent*; that is, all ways to start from point $\rho_1$ and end up in point $\rho_2$ should be equivalent, even if apparently distinct. The needed equivalences exactly correspond to the group laws, with closure internalized by the inductive definition of (3):

$$(\rho_1 +_{\mathbb{P}} \rho_2) +_{\mathbb{P}} \rho_3 = \rho_1 +_{\mathbb{P}} (\rho_2 +_{\mathbb{P}} \rho_3) \tag{L1}$$

$$\rho +_{\mathbb{P}} \mathbb{0} = \rho = \mathbb{0} +_{\mathbb{P}} \rho \tag{L2}$$

$$\rho +_{\mathbb{P}} (-\rho) = \mathbb{0} \tag{L3}$$

The (unsurprising) insight here is that paths in a sequence form a free group, generated by a single generator ($\mathbb{1}$) – the uniqueness of the generator exceptionally also makes the group abelian (*i.e.*, commutative). For convenience, we adopt the notational shorthand $\mathbb{1}^p$, where:

$$\mathbb{1}^p := \begin{cases} \underbrace{\mathbb{1} +_{\mathbb{P}} \cdots +_{\mathbb{P}} \mathbb{1}}_{p} & p \geq 0 \\ \underbrace{(-\mathbb{1}) +_{\mathbb{P}} \cdots +_{\mathbb{P}} (-\mathbb{1})}_{-p} & p < 0 \end{cases} \tag{4}$$

**Semantics**  The syntactic specifications of the previous paragraph impose constraints on the candidate semantic targets. Among these candidates, we isolate and focus on $\langle \mathbf{W} \rangle$, the subgroup of the orthogonal group $O(d)$ that is generated by a single orthogonal matrix $\mathbf{W}$. This semantics is not only sound[1] with respect to the structure under scrutiny, but also a familiar object in machine

---

[1]It is also complete except for the odd case where $\mathbf{W}^p = \mathbf{I}$ for some $p$. In practice, this kind of periodic behaviour does not arise randomly, and we can think of $\langle \mathbf{W} \rangle$ as being *isomorphic* to $\mathbb{P}$.

learning literature [Arjovsky et al., 2016, Bernardy and Lappin, 2022, *inter alia*]. Note that for $\langle \boldsymbol{W} \rangle$, the group axioms are obtained for free from the orthogonal group, and the additional requirement of commutativity is again satisfied by the uniqueness of the generator.

To illustrate the correspondence between the two structures (and at risk of being pedantic), we spell out the homomorphism $\lceil . \rceil$, which maps paths $\mathbb{P}$ to elements of $\langle \boldsymbol{W} \rangle$, and path operations to operations on orthogonal matrices of size $d$. For the primitives, we have $\lceil \mathbb{0} \rceil := \boldsymbol{I}_d$ and $\lceil \mathbb{1} \rceil := \boldsymbol{W}$. Path composition amounts to matrix multiplication, *i.e.*, $\lceil \rho_1 +_{\mathbb{P}} \rho_2 \rceil := \lceil \rho_1 \rceil \lceil \rho_2 \rceil$, while path inversion corresponds to matrix transposition, *i.e.*, $\lceil - \rho \rceil := \lceil \rho \rceil^{-1} \equiv \lceil \rho \rceil^{\top}$. The fact that orthogonal matrices form a group under multiplication is folklore; one can easily verify that the group laws hold also for the semantics.[2]

**Implementation** In practice, we have $\lceil \mathbb{1}^p \rceil \mapsto \boldsymbol{W}^p$; a norm-preserving bilinear form $\mathbb{R}^d \times \mathbb{R}^d \to \mathbb{R}$ which can be used to mediate the dot-product between a query $q$ and a key $k$ offset by a relative distance of $p$. The representation of all paths up to length $p$ can thus be implemented as a matrix collection $[\boldsymbol{W}^0, \dots, \boldsymbol{W}^p]$, which can asymptotically be obtained using $\mathcal{O}(\lceil \log_2(p) \rceil)$ matrix products (of exponentially larger matrices), and taking up the storage space equivalent of $(pd^2)$ floats. Transposed, the same matrices also serve to represent backwards paths $[\boldsymbol{W}^{-p}, \dots, \boldsymbol{W}^0]$. Storing the representations of all relative paths between queries and keys in a tensor $\boldsymbol{T} : \mathbb{R}^{m \times n \times d \times d}$, we may then substitute the dot-product term of equation (1) for the tensor contraction:

$$\sum_{\alpha,\beta} \boldsymbol{X}_{m\alpha} \boldsymbol{\Phi}_{\alpha\beta}^{(q)} \boldsymbol{T}_{mn\beta\gamma} \boldsymbol{Y}_{n\delta} \boldsymbol{\Phi}_{\delta\gamma}^{(k)} \tag{5}$$

Albeit transparent, this reduction strategy is computationally unappealing due to the doubly quadratic nature of $\boldsymbol{T}$. We can do better by noting that $\boldsymbol{T}_{mn}$ is (definitionally) equal to:

$$\boldsymbol{T}_{mn\alpha\beta} = \sum_{\gamma} \boldsymbol{A}_{m\gamma\alpha}^{(X)} \boldsymbol{A}_{n\gamma\beta}^{(Y)} \tag{6}$$

where $\boldsymbol{A}^{(X)}$ and $\boldsymbol{A}^{(Y)}$ are the matrices containing representations for the *absolute* positions of the entries in $\boldsymbol{X}$ and $\boldsymbol{Y}$, respectively. Concretely, a single relative representation is built by composing the *inverted* representation of the source with the representation of the target. Intuitively, each query follows the path that takes it *back* to the origin, which then allows it to directly combine with each forward-offset key; see Figure 1a for a visual example. This insight allows us to keep the memory footprint of equation (1) unchanged, replacing expression (5) with:

$$\sum_{\alpha,\beta,\gamma,\delta,\epsilon} \boldsymbol{X}_{m\alpha} \boldsymbol{\Phi}_{\alpha\beta}^{(q)} \boldsymbol{A}_{m\gamma\beta}^{(X)} \boldsymbol{A}_{n\gamma\delta}^{(Y)} \boldsymbol{Y}_{n\epsilon} \boldsymbol{\Phi}_{\epsilon\delta}^{(k)} \tag{7}$$

This version decomposes the tensor contraction into two matrix multiplications, essentially transforming (rotating or reflecting) the entries of $\boldsymbol{X}$ and $\boldsymbol{Y}$ independently according to their positions.

## 3.2 Intermezzo: Equivalence with RoPE

The story so far should be reminiscent of the rotary positional encoding scheme of Su et al. [2023, RoPE]. Not unlike our approach, RoPE substitutes the vanilla dot-product for a position-dependent bilinear form. Underlying the form is a $d \times d$-dimensional matrix $\boldsymbol{R}$ with a block-diagonal structure, where each $2 \times 2$-sized block corresponds to a rotation matrix that acts on a 2-dimensional subspace of $\mathbb{R}^d$. These independent rotations are parameterized by a (fixed) set of base angles $\Theta := [\theta_1, \dots, \theta_{d/2}]$. To incorporate position-dependence, *i.e.*, for a query/key pair at a relative distance of $p$, the base angles are multiplied by $p$, effectively altering the rotations applied.

At first glance, rotary encodings appear to be under-parameterized, and thus strictly weaker than orthogonal ones. However, any orthogonal matrix $\boldsymbol{W} \in O(d)$ admits a canonical form $\boldsymbol{W} = \boldsymbol{P}\boldsymbol{Q}\boldsymbol{P}^{\top}$, where $\boldsymbol{P}$ is an orthogonal change of basis, and $\boldsymbol{Q}$ is block-diagonal, with the $2 \times 2$-sized blocks being, once again, $2-$dimensional rotation matrices [Murnaghan and Wintner, 1931][3]. Owing to the orthogonality of $\boldsymbol{P}$, raising $\boldsymbol{W}$ to its $p$th power is equal to $\boldsymbol{P}\boldsymbol{Q}^p\boldsymbol{P}^{\top}$ (*i.e.*, it leaves the change of basis unaffected). In turn, raising $\boldsymbol{Q}$ to its $p$th power is equivalent to simply multiplying the rotation

---

[2]The story is no different for $\boldsymbol{W}$ unitary, with the group structure provided by the unitary group $U(d)$, and path inversion interpreted as the matrix conjugate transpose.

[3]We alert the reader that a *constructive* proof of this decomposition has proven surprisingly difficult to find.

angles of its blocks by $p$. Finally, given the linearity of the transformations $\mathbf{\Phi}^{(q)}$ and $\mathbf{\Phi}^{(k)}$, their compositions with $\boldsymbol{P}$ are also linear. By identifying $\boldsymbol{Q}$ with RoPE's $\boldsymbol{R}$, we can then see that, for any given collection of angles $\Theta$, APE and RoPE coincide under the substitutions:

$$\mathbf{\Phi}^{(q)}_{\text{RoPE}} = \mathbf{\Phi}^{(q)}\boldsymbol{Q} \quad \text{and} \quad \mathbf{\Phi}^{(k)}_{\text{RoPE}} = \mathbf{\Phi}^{(k)}\boldsymbol{Q} \tag{8}$$

In practical terms, and uniquely for the sequential case, APE *is equivalent to a trainable version of RoPE, where the rotation angles $\Theta$ may vary and be optimized during training.*[4]

Which of the two parameterizations is preferable is debatable. On the one hand, APE's formulation is FLOP-optimized (being just matrix multiplications), and obviates the need for backpropagating through trigonometric functions (which are periodic, non-monotonic, and prone to gradient instabilities). On the other hand, RoPE's diagonalized form gives access to a memory-efficient contraction that does away with the matrix multiplications of expression (7) altogether; we direct the interested reader to Su et al. [2023, Section 3.4.2] for a reference implementation.[5]

In either case, the equivalence between the two is confined to the *sequential* setup; we will now move on to generalize our strategy to other, *previously inaccessible*, structures.

### 3.3 Trees

**Syntax** In the previous section, we characterized the structure of relative paths on a sequence as the free group with one generator, and uncovered a (practically) isomorphic interpretation in the subgroup of orthogonal matrices with a single generator. Upon closer inspection, we note that a sequence can be viewed as a special case of the more general structure of $\kappa$-ary branching trees, where the branching factor $\kappa$ just so happens to be 1. Denoting the more general case as $\mathbb{P}_\kappa$, we must first extend the set of primitives to include all branching options, $\mathbb{1}, \mathbb{2}, \ldots \kappa : \mathbb{P}_\kappa$. Each primitive now denotes a choice of branch (except for $\mathbb{0}$, which is again the empty path). Paths now form a free group with $\kappa$ distinct generators. The presence of multiple generators means that commutativity no longer holds; $\mathbb{1} +_{\mathbb{P}_\kappa} \mathbb{2}$ is distinct from $\mathbb{2} +_{\mathbb{P}_\kappa} \mathbb{1}$ (the former prescribes a descent down branch $\mathbb{1}$ then branch $\mathbb{2}$, whereas the latter prescribes a descent down branch $\mathbb{2}$ then branch $\mathbb{1}$). Inversion is as before: for every path from each local root to some descendant down the line, there is also an inverse path from that descendant up to its ancestor. Perhaps more interestingly, upwards and downwards paths can be joined, allowing the precise specification of relative paths between any two nodes, even when the two do not share a single line of descent (think nephews, aunts and all other sorts of distant relatives, see Figure 1b for an example). Adjusting grammar (3) accordingly, we have:

$$\mathbb{P}_\kappa := \mathbb{0} \mid \mathbb{1} \mid \mathbb{2} \mid \ldots \mid \kappa \mid \mathbb{P}_\kappa +_{\mathbb{P}_\kappa} \mathbb{P}_\kappa \mid -\mathbb{P}_\kappa \tag{9}$$

with laws L1, L2 and L3 still in effect.

**Semantics** The interpretation follows along the same lines as before. This time around, however, we cannot make do with a single orthogonal matrix $\boldsymbol{W}$ – we need a collection of $\kappa$ matrices, one for each branch option. As a consequence, the semantic target is now $\langle \boldsymbol{W}_1, \boldsymbol{W}_2, \ldots \boldsymbol{W}_\kappa \rangle$. Note that the target is no longer commutative (exactly in alignment with the source).

**Implementation** For a tree structure of depth $\delta$ and branching factor $\kappa$, let $\nu$ denote the number of *unique* absolute positions occupied (upper bound by $\kappa^\delta$ in the case of a complete tree). Their representations can be computed in $\delta\kappa$ steps of parallel matrix-matrix multiplications and a memory cost of $\nu d^2$, as follows. First, we can build up a collection of all unique absolute paths, each represented as a (right-padded) word of length $\delta$ from the vocabulary of primitives. Their corresponding representations constitute a tensor of size $\nu \times d \times d$, initialized as $\nu$ identity matrices. We can then iterate across these words in parallel, one primitive per step (*i.e.*, depth) $t$, selecting all words that take the same branching direction at the current depth, and right-multiplying their representations by the corresponding orthogonal generator. Finally, absolute paths can be composed into relative ones using the modified dot-product attention of expression (7), just like before.

---

[4]An alternative reading is that even though orthogonal matrices are generally more expressive than rotation matrices (allowing not just rotations but also reflections), the Transformer's architecture makes up for RoPE's reduced expressivity by supplying a free change of basis through its trainable weights $\mathbf{\Phi}$.

[5]For more practical insights on initializing and parameterizing APE and translating between APE and RoPE, please refer to Appendix A.

## 3.4 Grids

The generalization from sequences to trees rests on the observation that a sequence is a tree with a deficit of choices. An altogether different axis of generalization can be obtained by recalling that composite groups can be constructed by joining together two or more elementary groups. Moreover, if it just so happens that the original groups were abelian, then so is their composition; in that case, we call the composite a *group direct sum*. This construction provides access to an extension from sequences to multidimensional regular grids.

For the sake of simplicity and without loss of generality, we consider a standard instance of a two-dimensional grid: an image. An image is a collection of pixels (or pixel patches) that inhabit a coordinate system $(h, w)$. Each of $h$ and $w$ is the product of grammar (3), inheriting all path-related notions discussed earlier. Since $\mathbb{P}$ is an abelian group, the coordinate system also constitutes an abelian group $\mathbb{P}^2 := \mathbb{P} \oplus \mathbb{P}$. The new group and inversion operations are $+_{\mathbb{P}^2}$ and $(-)_{\mathbb{P}^2}$, and denote the act of joining and inverting two-dimensional paths, respectively. Both are canonically defined component-wise, on the basis of their one-dimensional counterparts:

$$(x, y) +_{\mathbb{P}^2} (z, w) := (x +_{\mathbb{P}} y, z +_{\mathbb{P}} w) \tag{10}$$

$$-(x, y) := (-x, -y) \tag{11}$$

with $\mathbb{0}^2 := (\mathbb{0}, \mathbb{0})$ as the new neutral element. Intuitively, $+_{\mathbb{P}^2}$ corresponds to vector addition, and $(-)_{\mathbb{P}^2}$ to a reflection about the origin with respect to both axes.

**Semantics**    The specifications above allow us to reuse the notions from Section 3.1 in order to interpret the components and operations of $\mathbb{P}^2$. What is left unspecified is the interpretation of the group elements themselves; that is, we have yet to explicate what an object of $\lceil \mathbb{P} \oplus \mathbb{P} \rceil$ looks like. The quest is a short one; the notion of a direct sum carries over to matrices, and is defined as:

$$\boldsymbol{A} \oplus \boldsymbol{B} := \begin{bmatrix} \boldsymbol{A} & \boldsymbol{0} \\ \boldsymbol{0} & \boldsymbol{B} \end{bmatrix} \tag{12}$$

From this, we get the (rather straightforward) interpretation $\lceil (\rho_1, \rho_2) \rceil \mapsto \lceil \rho_1 \rceil \oplus \lceil \rho_2 \rceil$.

**Implementation**    In practice, we now split the vector space in two independent parts. The first part is modulated by orthogonal matrices from $\langle \boldsymbol{H} \rangle$, and the second part by orthogonal matrices from $\langle \boldsymbol{W} \rangle$. For a query $q$ and a key $k$ that reside at a relative distance of $(h, w)$, their attention score is computed as $q(\boldsymbol{H}^h \oplus \boldsymbol{W}^w)k$ – see Figure 1c for an illustration. Each axis contributes an additive but separable factor to the attention score, forcing the model to learn contextual alignments between token pairs on the basis of their coordinate-wise distances. Not much else is different: we can still compute all matrices in parallel, temporally bound by a logarithmic complexity of $\log_2(\max(h, w))$ and $\max(h, w)(\frac{d}{2})^2$ storage space, given a grid of size $(h, w)$. Subquadratic memory complexity can once more be achieved by virtue of diagonalization, just as in the sequential case.

## 3.5 Variants & Extensions

The structures that we have seen so far are not the only ones that our methodology can tackle – in fact, many other group-like structures are amenable to similar interpretations. We sketch out some enticing examples below.

**Absolute Positions**    Our analysis has so far focused on paths *relative* to positions. Fixing the point of origin allows a straightforward simplification to *absolute* positions. The new structure is that of a *monoid*: there's no longer an inversion, and laws L1 and L2 only are now in effect. The framework remains largely unchanged: one can still use subgroups of matrices to represent positions, except this time applying them on either the queries or the keys (rather than both).

**Periodic Domains**    Under addition, the integers form an *infinite* cyclic group. An interesting twist would be to consider the positional encodings of *finite* cyclic groups instead. Such structures are not uncommon; in chemistry, for instance, a benzene molecule comprises six carbon atoms arranged in a ring. The semantics of such a structure would need to be of a matching period; that is, we would need a generator $\boldsymbol{W}$ such that $\boldsymbol{W}^6 = \boldsymbol{I}$. Such a parameterization is straightforward; we simply need to fix the orthogonal matrix so as to have it implement rotations at angle-multiples of $\pi/3$.

**Time Series & Subsampling**    Our sequential case analysis assumed a dense sequence with a uniform sampling rate. However, our strategy also applies to any series, even if sparsely sampled, as long as the sampling rate is quantized (*i.e.*, a multiple of some constant step). That is, positional indices (and their representations) do not need to match the placement of tokens in the sequence.

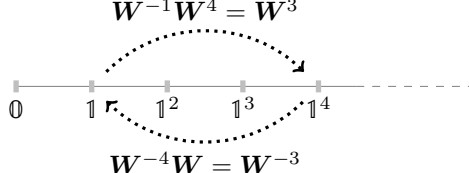

(a) The half-axis of absolute positions on a sequence, with a visualization of the two directions of relative paths between points 1 and 4. In either case, the interpretation is the matrix multiplication of the inverted source against the target.

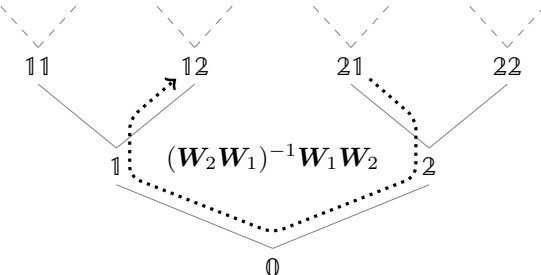

(b) The space of paths on binary branching trees, with an illustration of the relative path from $21$ to $12$. Same as before, the interpretation is the matrix multiplication of the inverted source against the target

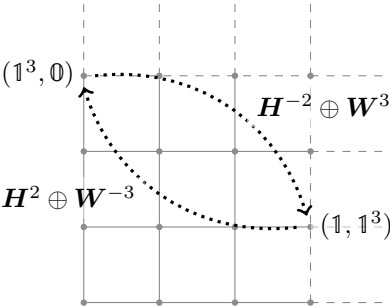

(c) The quarter-plane of absolute positions on a 2-dimensional grid, with a visualization of the two directions of relative paths between points $(3, 0)$ and $(1, 3)$. The interpretation is now a block-diagonal matrix consisting of the blocks interpreting the path over each coordinate.

Figure 1: Example paths and their interpretations across the structures examined.

**Composite Groups** The direct sum interpretation of Section 3.4 is applicable for arbitrary groups that can be described as products, commutative or otherwise. This allows the representation of positional encodings for several other kinds of composite structures that can be concocted using the same principles, such as sequences of trees, trees of grids, etc.

**Beyond Dot-Product Attention** Throughout the previous sections, we have adopted a dot-product formulation for the attention weight function. Nonetheless, APE can be readily integrated into any other attention mechanism, such as linear [Katharopoulos et al., 2020], cluster [Vyas et al., 2020] and "softmax-free" [Lu et al., 2021] variants, *inter alia*.

## 4  Experiments

To assess the viability of our approach, we conduct a series of experiments across a range of tasks, in setups that allow for replicable and reliable comparisons with alternatives. When using APE, we follow Wu et al. [2021] in scaling the dot-product score between two tokens at a distance of $p$ (*i.e.*, $p$ steps away) by $p^c$; here, we set $c := 0.98$. This serves to stabilize training by introducing a locality bias (or long-distance decay) factor. For the sake of parameter compression, we share the orthogonal matrices between the different encoder/decoder layers, but use a distinct matrix (or collection of matrices) per head. To isolate and quantify the effect of initialization, we report results on two

different initialization strategies: one where the orthogonal operators are set to mimic RoPE rotations (default), and one where they are set to be close to the identity (no init). Similarly, to isolate and quantify the effect of trainability when comparing to RoPE, we report results over both fixed (frozen) and trainable (tuned) rotation angles.

We provide an extensive account of our experimental setups in Appendix B.

## 4.1 Sequence Transduction

**Machine Translation**    First, we follow Vaswani et al. [2017] in training a Transformer$_{\text{BASE}}$ model on machine translation over WMT14 EN→DE [Bojar et al., 2014].

To provide a comprehensive comparison, we pit our proposed methodology against standard positional encoding schemes from the literature: the vanilla *Sinusoidal* encodings of Vaswani et al. [2017], the *Absolute* encodings of Gehring et al. [2017], the *Relative* encodings of Shaw et al. [2018] and the *Rotary* encodings of Su et al. [2023]. To ensure a fair comparison, we allow all models the exact same budgets (both memory and time).

**Synthetic Tasks**    We further examine three standard sequence transduction tasks: sequence copying, sequence reversal, and sequence repetition. These are meant to directly assess each model's capacity for algorithmic induction, in setups where explicit position-based addressing, both absolute and relative, is required.

## 4.2 Tree Transduction

Next, we consider four algorithmic transduction tasks on binary branching trees: tree copying, recursive tree rotation up to a fixpoint, algebraic reduction of $C_3$ expressions, and self-referential tree manipulation; see Appendix B for details.

In addition to previous sequential baselines, we compare our model to the encodings of Shiv and Quirk [2019, *Tree-SQ*]. For all four tasks, we experiment with both breadth-first and depth-first decoding.

## 4.3 Image Recognition

Finaly, we train a Compact Convolutional Transformer [Hassani et al., 2021] on CIFAR-10 [Krizhevsky et al., 2009].

Typically, attention-based architectures for vision rely on additive positional encoding schemes, applied on the image prior to it being sequentialized (row-by-row flattened). Here, we compare fixed [Wang and Liu, 2019, *Sinusoidal 2D*] and parametric [Gehring et al., 2017, *Absolute*] variants of the above against both the sequential and the grid-structured versions of our scheme.

## 4.4 Results

We repeat each experiment three times, varying the seeds used for weight initialization and optimization, but fixing the data across repetitions. We report means and 95% CIs in Table 2. We highlight each category's best (in green), and underline scores where the CI spans the mean of the respective best.

At the macro level and consistently across modalities, domain-appropriate algebraic interpretations match or surpass strong and specialized baselines – without *any* hyper-parameter tuning or search. Specifically, across the 13 setups considered, APE is the uncontested top performer in 8, ranks among the best in 3, and falls within the confidence margin of the top performer in one. Exceptionally, in the breadth-first version of the tree-copy task, tree algebraic encodings are surpassed by a handful of sequential alternatives; this is no surprise, since in this case the tree structure is practically a task-irrelevant syntactic confound. Perhaps more surprisingly, in the breadth-first version of the tree-manipulation task, tree algebraic encodings are surpassed only by their non-initialized, sequential version; an anomaly likely due to a single repetition with an unusually low perplexity score.

We also note three general trends. First, initializing APE to match RoPE frequency bands at the start of training consistently and significantly improves performance, possibly because RoPE rotary primitives have undergone empirical tuning for stability and performance. Second, given identical initialization, a sequential APE generally outperforms a trainable RoPE, despite their theoretical equivalence. This might be due to the difficulty of optimizing periodic signals (*i.e.*, RoPE's trigonometric functions) compared to APE's (orthogonal) matrix multiplications. Third, a frozen RoPE performs comparably to

| | Scheme | | | | | | |
|---|---|---|---|---|---|---|---|
| Task | *Sinusoidal* | *Absolute* | *Relative* | *Rotary* *(frozen)* | *(tuned)* | *Algebraic* *(/w init)* | *(w/o init)* |
| WMT14 EN→DE (BLEU / ↑) | 14.57±0.12 | 22.09±0.11 | 23.15±0.03 | 24.03±0.06 | 23.92±0.20 | 23.93±0.10 | 23.84±0.10 |
| COPY | 1.01±0.00 | 1.11±0.00 | 1.00±0.00 | 1.00±0.00 | 1.00±0.00 | 1.00±0.00 | 1.00±0.00 |
| REPEAT | 1.85±0.15 | 3.66±0.06 | 1.44±0.16 | 1.08±0.12 | 1.00±0.00 | 1.00±0.00 | 1.02±0.00 |
| REVERSE (PPL. / ↓) | 3.92±0.99 | 4.62±0.67 | 4.08±1.12 | 1.09±0.02 | 1.01±0.00 | 1.01±0.00 | 1.03±0.02 |

(a) Performance results on neural machine translation and synthetic sequence transduction.

| | Task/Regression | | | | | | | |
|---|---|---|---|---|---|---|---|---|
| | COPY | | ROTATE | | C₃ | | TREE-OPS | |
| Scheme | breadth | depth | breadth | depth | breadth | depth | breadth | depth |
| *Sinusoidal* | 1.06±0.01 | 5.68±0.63 | 6.93±0.38 | 7.13±0.35 | 2.66±0.10 | 2.78±0.08 | 20.53±7.11 | 64.86±6.41 |
| *Tree-SQ* | 1.29±0.01 | 1.07±0.00 | 2.60±0.16 | 1.87±0.24 | 2.27±0.59 | 2.29±0.24 | 19.18±3.23 | 16.41±6.14 |
| *Absolute* | 6.64±0.12 | 7.02±0.17 | 7.77±0.15 | 7.24±0.20 | 2.77±0.21 | 2.79±0.22 | 37.78±0.72 | 48.91±5.83 |
| *Relative* | 1.01±0.00 | 6.12±0.06 | 6.00±0.25 | 7.72±0.28 | 1.70±0.07 | 2.43±0.04 | 2.36±0.02 | 16.86±1.27 |
| *Rotary (frozen)* | 1.42±0.58 | 2.46±0.59 | 4.58±0.30 | 4.97±1.79 | 1.55±0.34 | 2.15±0.22 | 2.53±0.08 | 33.54±9.04 |
| *Rotary (tuned)* | 1.00±0.00 | 1.70±0.05 | 4.07±0.34 | 2.60±0.11 | 1.08±0.02 | 1.90±0.22 | 2.55±0.05 | 20.87±0.33 |
| *Algebraic (**seq**)* | 1.00±0.00 | 1.63±0.06 | 2.95±0.08 | 2.48±0.27 | 1.07±0.01 | 1.83±0.02 | 2.30±0.03 | 20.05±0.36 |
| *w/o init* | 1.00±0.00 | 2.36±0.63 | 5.18±0.10 | 5.72±1.23 | 1.45±0.08 | 2.29±0.06 | 1.75±0.74 | 29.26±9.15 |
| *Algebraic (**tree**)* | 1.01±0.00 | 1.00±0.00 | 1.05±0.01 | 1.01±0.00 | 1.00±0.00 | 1.00±0.00 | 2.24±0.06 | 1.83±0.02 |
| *w/o init* (PPL. / ↓) | 1.07±0.00 | 1.04±0.08 | 1.44±0.15 | 1.27±0.15 | 1.05±0.10 | 1.00±0.00 | 2.42±0.01 | 1.86±0.01 |

(b) Performance results on algorithmic tree manipulation tasks.

| | Epoch | |
|---|---|---|
| Scheme | ≤150 | ≤300 |
| *Sinusoidal 2D* | 91.57±0.01 | 92.79±0.20 |
| *Absolute* | 90.86±0.19 | 92.68±0.39 |
| *Algebraic (**seq**)* | 92.68±0.24 | 94.59±0.15 |
| *w/o init* | 88.93±0.19 | 91.09±0.20 |
| *Algebraic (**grid**)* | 93.13±0.33 | 94.67±0.06 |
| *w/o init* (ACC. / ↑) | 92.95±0.07 | 94.48±0.18 |

(c) Best-by-epoch top-1 accuracy scores on image recognition on CIFAR-10.

Table 2: Experimental results and baselines across the tasks considered.

a randomly initialized APE in most tasks considered, suggesting that adjusting rotoreflection angles during training is not necessarily better than adjusting rotation planes while keeping the angles fixed. Contrary to all the above, a frozen ROPE weakly outperforms both a tunable ROPE and an initialized APE in the neural machine translation task; likely an artifact of attention overfitting to specific positional patterns.

## 5   Related Work

Dense attention is by now a foundational component of various problem- and domain-general architectures. Combined with its structural indifference, this underscores the pressing need for learning strategies capable of injecting structural biases directly at the representation level. As such, positional encodings have garnered significant community attention in recent years – too much, in fact, to permit an exhaustive enumeration here. An extensive survey and meta-review is provided by Dufter et al. [2022] who group and rank these works on the basis of several criteria. Our work presents a universal, intuitive and formally grounded recipe that meets *all* these criteria: it is *trainable*, amenable to problem-specific and data-driven tuning; *reference-adjustable*, allowing both absolute and relative positional specifications; *unbounded*, capable of representing enumerably infinite positions irrespective of model instantiation and/or the targeted data size; *contextual*, implementing a dynamic effect that varies depending on token content; *effective*, consistently matching or surpassing

baselines in the tasks considered; and, finally, *efficient*, exhibiting generally favorable asymptotic complexities.

We must point out that the concept of positional encodings as sequence homomorphisms has already been hinted at, first by Wang et al. [2020] and later by Su et al. [2023], even if not explicitly formulated as such. Despite approaching the problem from different angles, both approaches interpret positions as multiplicative, norm-preserving (rotation-like) operations. Our proposal expands upon these two, first in providing a proper algebraic framing of the problem, and second in extending the interpretation from rotations around the axes to rotations and reflections about arbitrary planes. In the case of a single generator matrix (*i.e.*, sequences), this difference turns to be non-essential, being practically neutralized by the Transformer's trainable weights. This no longer holds, however, in the case of multiple generator matrices (*i.e.*, grids or trees), where each generator should be able to rotate and reflect different sets of planes. In that sense, algebraic positional encodings offer an appealing unifying perspective of a multidimensional generalization to the aforementioned rotation-based frameworks. This sentiment is shared by Lim et al. [2023] who, in parallel to our work, similarly advocate for positional encodings as group homomorphisms, there framed as irreducible group representations. Modulo presentation, the two approaches are variations on a common theme; theirs is technically concerned with post-hoc representation of symmetries and equivariances at a per-datum scale, whereas ours focuses on the interpretation of domain signatures at the dataset scale.

More generally, algebraic manipulations are not uncommon in modern machine learning literature. The recognition of abstract algebra as a practical tool for imposing structural well-behavedness has led to its increased adoption as the go-to recipe for structure-informed neural architectures, largely obsoleting the inefficient and *ad hoc* augmentation routines of the past. This line of work can be traced back to the group equivariant convolutions of Cohen and Welling [2016], which have by now bloomed into a field of their own; see Weiler et al. [2023] for an up-to-date overview.

## 6 Limitations

We recognize weaknesses and limitations across three fronts. On the *theoretical* front, we have limited our scope to simple inductive groups, consciously ignoring potential interpretations of more complex constructions. We defer this to future work. On the *empirical* front, having to recompute positional encodings once per batch increases a model's temporal complexity during training. While this is barely noticeable in sequential and grid constructions, which scale logarithmically, it becomes evident when dealing with complete trees, which scale linearly and require explicit for-loops. On the *epistemic* front, we conducted a limited set of experiments, focusing primarily on replicability and fairness. We leave more exhaustive empirical comparisons on practical downstream tasks to future work or interested parties.

## 7 Conclusion

We have presented a theoretically motivated approach towards constructing positional encodings for a variety of structures. Without any significant modification or overhead, our methodology can capture sequences and their (multi-dimensional as well as multi-branching) generalizations. In doing so, it reconciles powerful but structurally oblivious models with their missing inductive biases, permitting structure-aware architectural refinements across a range of tasks and setups (see also Kogkalidis et al. [2024] for parallel work employing the methodology in a neurosymbolic representation learning setup). Beyond that, our approach grants full control over how these biases are to be implemented, while also being amenable to adjustments and extensions. Our work indicates that generality and extensibility are not *in spite of*, but rather *due to* structural discipline and abstraction. We perceive it as an important step towards data-efficient, general, and transparent models of neural computation.

## Acknowledgments and Disclosure of Funding

KK and VG were supported by Saab-WASP via the project "Neurodynamic Programming and Reinforcement Learning" (grant 411025). VG also acknowledges the support from Academy of Finland (grant 342077) for "Human-steered next-generation machine learning for reviving drug design", and the Jane and Aatos Erkko Foundation (grant 7001703) for "Biodesign: Use of artificial intelligence in enzyme design for synthetic biology".

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

# A  Parameterizing APE

## A.1  Orthogonalization

The orthogonal primitives underlying APE can be procured by matrix-exponentiating skew-symmetric bases. Concretely, for some cyclic group $\langle C \rangle$:

1. Start with an *upper triangular* matrix $A$; this matrix parameterizes the entire group.
2. Obtain the *skew symmetric* $B := A - A^\top$
3. Obtain the *matrix exponent* $C := \mathrm{expm}(B)$; the resulting matrix is *orthogonal*, and acts as the group's generator.

## A.2  Switching between APE and RoPE

In the commutative (direct sum of finitely many cyclic groups) case, it is possible to switch freely between APE and RoPE. Doing so might be useful, *e.g.*, for initializing APE, for inspecting the learned rotoreflections post-training, or for making use of RoPE's memory-optimized vector-multiplication formula in a system originally trained with APE. Note that here we consider the purely real-valued version of RoPE (and APE).

**RoPE $\to$ APE**   To convert RoPE to APE for some collection of angles $\Theta := [\theta_1, \ldots \theta_n]$:

1. Expand $\Theta$ into a rotation matrix $C$,

$$
C := \begin{bmatrix}
cos\theta_1 & -sin\theta_1 & 0 & 0 & \ldots \\
sin\theta_1 & cos\theta_1 & 0 & 0 & \ldots \\
0 & 0 & cos\theta_2 & -sin\theta_2 & \ldots \\
0 & 0 & sin\theta_2 & cos\theta_2 & \ldots \\
\vdots & \vdots & \vdots & \vdots & \ddots
\end{bmatrix}
$$

   **Note**: Stop here if not interested in parameterizing $C$.
2. Use a solver to approximate the *matrix logarithm* of $C$, $B := \mathrm{logm}(C)$.
3. Find a matrix $A$ such that $\mathrm{mse}(B, A - A^\top) \le \epsilon$, *e.g.*, using a numerical optimizer. Matrix $A$ can be used to parameterize the group, *cf.* A.1.

**APE $\to$ RoPE**   To convert APE to RoPE for some cyclic group $\langle W \rangle$:

1. Find the normal form $W = PQP^\top$.
2. Extract the angles in each block of $Q$; the resulting collection of angles is RoPE's $\Theta$.
3. For each attention head involved, right-compose the Transformer's $\Phi^{(q)}$ and $\Phi^{(k)}$ with $P$.

# B  Experimental Setups

## B.1  Machine Translation

For our machine translation experiments, we use the official dataset breakdown (including the extended evaluation set). We tokenize the training and evaluation sets with MOSES, using the standard pipeline: punctuation normalization $\to$ unicode normalization $\to$ language-specific tokenization.[6] We apply byte-pair encoding [Gage, 1994, Sennrich et al., 2016] using the subword-nmt package.[7] We apply 32k merges across the source and target training corpora, without truncating the resulting (shared) vocabulary (of size 35 533). Our loss term is given as the cross-entropy between the teacher-forced predictions and the ground-true labels, smoothed by 10%. We train in a distributed environment consisting of 4 GPUs, with a batch size of 3 072 target tokens per GPU. We average gradients and update parameters once every 2 GPU iterations (or: 8 batches). We optimize using Adam with a learning rate dictated by the schedule prescribed by Vaswani et al. [2017]. We stop optimizing after 150 000 parameter updates or 16 hours, whichever comes first. Throughout training, we circularly store the 10 best checkpoints, ranked on the basis of dev set loss (evaluated once every 500 updates). During inference, we average the 10 checkpoints into a single model, and select hypotheses from a beam of width 4 and a length penalty of 0.6 [Wu et al., 2016]. We report BLEU scores over the *test set* (newstest2014), comparing the BPE-merged and detokenized output against the raw references using sacrebleu [Post, 2018].[8]

---

[6]See https://github.com/moses-smt/mosesdecoder

[7]See https://github.com/rsennrich/subword-nmt.

[8]Signature: nrefs:1 | case:lc | eff:no | tok:13a | smooth:exp | version:2.4.2.

|  | Experiment/Value | | |
| Parameter | NMT | Transduction | Image |
| --- | --- | --- | --- |
| Convolution Size | – | – | (3,3) |
| Convolution Stride | – | – | 1 |
| Embedding Size | 512 | 512 | 256 |
| Feedforward Size (enc) | 2048 | 512 | 512 |
| Feedforward Size (dec) | 2048 | 1024 | – |
| Feedforward Activation | ReLU | ReLU | GELU |
| # Layers (enc, dec) | (6, 6) | (2,2) | (7, 0) |
| # Heads | 8 | 8 | 4 |
| Norm | LayerNorm | LayerNorm | LayerNorm |
| Norm Position | Post | Pre | Pre |

Table 3: Hyperparameter setups, grouped by experiment.

## B.2 Synthetic Transduction

**Tree Task Descriptions** The tree copy task is morally identical to its sequential version – the tree structure (and its positional specification) is practically a confound.

In the tree rotation$^\star$ task, the output tree is the result of recursively right-rotating all subtrees of the input. The task is challenging but purely structural, in the sense that its resolution requires no real interaction between content and position.

For the algebraic expression reduction task, we consider input trees that specify a complex expression from the cyclic group $C_3$, and task the model with producing the result of a single reduction step (*i.e.*, reducing all subtrees of depth 1 into a leaf). This time around, the model has to identify reducible subtrees, match operators to their argument and collapse the three into a single node depending on their content.

The tree operations task, finally, combines the aspects of the other three, requiring content-based addressing, structure manipulation and dynamic semantics resolution. Concretely, we generate an input tree consisting of unique nodes, and randomly select one of its subtrees as well as one of four operators. We then construct a deeper tree, where the new root corresponds to the chosen operator, its left branch corresponds to the numerical index of the selected subtree, and the right branch is the original tree in its entirety. The model is then tasked with producing the correct output given this combination of an operator, a tree, and an index. We consider four operations: extraction (*i.e.*, return the indexed subtree), flip-extraction (*i.e.*, return the indexed subtree, rotated), truncation (*i.e.*, return the full tree with the indexed subtree removed) and a no-op (*i.e.*, return the full tree as-is, ignoring indexing).

**Hyperparameters** For all synthetic tasks, we generate disjoint train, dev and test sets of sizes 6 000, 2 000 and 2 000. We train a small Transformer model, optimizing with AdamW [Loshchilov and Hutter, 2017] for 400 epochs and a batch size of 64, using a linear warmup – cosine decay schedule. For the sequential tasks, we populate the datasets with words of random lengths from $\mathcal{N}(100, 10)$ and a vocabulary size of 20 (to ensure token repetition and diffuse the possibility for leaning on content-based addressing). For the tree tasks, we populate the datasets with non-uniform trees of random depths sampled from $\mathcal{N}(7, 1)$. For the tree-ops task, exceptionally, we set the vocabulary size to 128 so as to have enough unique nodes to allow content-based addressing.

When using a positional encoding scheme that requires fixing the size of the structure being modeled (*i.e.*, the *Tree*, *Relative*, and *Absolute* schemes), we fix it at approximately the maximum training size, practically ensuring the most stringent comparison.

In all experiments, we share source and target embedding weights between both the encoder-decoder embedding layers, and the decoder's classification head.

## B.3 Image Recognition

For our image recognition experiments, we largely rely on the setup of Hassani et al. [2021]. Concretely, we apply a small-step "tokenizing" convolution on the input image, downsample the result with max pooling and flatten the result into a sequence. After we pass the sequence through the encoder, we apply a global soft attention [Li et al., 2016, *inter alia*] (rediscovered by Hassani et al. [2021], there dubbed "sequence pooling") to aggregate into a single vector prior to applying the classifier. To attain competitive scores, we apply standard CIFAR-10 data augmentations and

more aggressive regularization: a 10% attention weight dropout, a stochastic depth of 10% for each consecutive layer, and a weight decay of $3 \cdot 10^{-2}$. The above settings and the hyperparameter setup are taken without modification from Hassani et al. [2021].

