# OpenReview forum: "Algebraic Positional Encodings"
_NeurIPS.cc/2024/Conference — NeurIPS 2024 spotlight_

### Official Review · Reviewer_PGKE · 2024-06-25

**Soundness:** 3
**Presentation:** 3
**Contribution:** 3
**Rating:** 6
**Confidence:** 4

**Summary:**

The paper presents a groundbreaking positional encoding strategy named  Algebraic Positional Encodings for Transformer-style models, The paper offers a versatile mapping technique that translates the algebraic specifications of a domain into orthogonal operators. By doing so, it preserves the fundamental algebraic characteristics of the source domain, ensuring that the model maintains its desired structural properties. The author also conducts experiments to validate the performance of Algebraic Positional Encodings.

--------------------------------------------------------------------------------------------------------------------------------------------------------------------------
Update 08/Aug/2024: I increase the score from 5 to 6 and the confidence from 2 to 3, after checking author rebuttal.

--------------------------------------------------------------------------------------------------------------------------------------------------------------------------
Update 08/Aug/2024: I increase the confidence from 3 to 4, after checking experiments of trainable RoPE.

**Strengths:**

* The paper proposes Algebraic Positional Encodings, which provide a flexible mapping from the algebraic specification of a domain to an interpretation as orthogonal operators.
* The paper has experiment results to prove the effectiveness of Algebraic Positional Encodings

**Weaknesses:**

* The paper is relatively hard to understand. If possible, could the author provide the pseudo-code or python code of the proposed Algebraic Positional Encodings? Or is there any other way that may help the reader to understand the implementation of the method?

* Why Algebraic (grid) performance is often better than Algebraic (seq)? If possible, could the author explain it?

* What is the length extrapolation performance of Algebraic Positional Encodings? For example, what is the performance of Algebraic Positional Encodings when the training length is smaller than the validation length, such as when training on length 128 and testing on length 256? One potential code base is https://github.com/google-deepmind/randomized_positional_encodings/tree/main. And author may also choose other potential base, as long as it can conduct the length extrapolation experiments.

**Questions:**

please see the above weakness.

**Limitations:**

Yes, the authors have discussed the limitations.

---

> ### Author Rebuttal · Authors · 2024-08-04
>
> Warm greetings, and thank you for your review.
>
> ---
>
> > `The paper is relatively hard to understand.`
>
> We are very sorry to hear. To summarize, what we we are basically saying is that:
> 1. The properties of several common I/O structures in ML literature can be perfectly described using elementary group theory.
> 2. Group theory offers insights on how to represent elements of these structures as matrices and linear maps.
> 3. Using these representations as general recipe for structure-aware positional encodings, we arrive at various generalizations of RoPE, which is the mostly widely adopted strategy and the current state of the art.
> 4. Our results suggest that indeed this seems to be working not just in theory, but also in practice.
>
> What exactly gave you a hard time? Is there something we can perhaps clarify?
>
> > `could the author provide the pseudo-code or python code`
>
> We have already made **all of our code** available for review. Please check the supplementary material. We'd be happy to provide clarifications if/as needed.
>
> > `Why Algebraic (grid) performance is often better than Algebraic (seq)?`
>
> Algebraic (grid) treats the image as a grid of distinct ($x$, $y$) positions, where up/down and left/right transitions are kept semantically separate. Algebraic (seq) flattens the image into a sequence, which obfuscates the semantics: a right(/left) transition could signify **either** the immediately adjacent pixel to the right(/left), **or** the first(/last) pixel of the next(/previous) image row. This is **not a weakness** but the **intended behavior**; we are precisely arguing for structure-aware positional encoding schemes.
>
> > `... length extrapolation ...`
>
> Great question! We don't expect astonishing extrapolation results out-of-the-box. That said, all insights that apply for RoPE should carry over to our approach -- see for instance *rerope*, *yarn*, *position interpolation*, *etc*. For the sake of transparency, we present length extrapolation results for the synthetic sequence tasks below. We do training and model selection on sequences of length 100±7, and evaluate on sequences of length 200±7. We report PPL scores (so less is better).
>
> | Task     | Sinusoidal | Absolute | Relative | Rotary | Algebraic |
> | ---------|------------|----------|----------|--------|---------|
> | Copy     |  18.9      | 22.3     | 1.0      | 1.0    | 1.0     |
> | Reverse  |  18.3	| 18.3	   | 4.5      | 14.6   | 4.8     |
> | Repeat   |  18.1      | 22.2     | 21.8     | 1182.7 | 262.3   |
>
> We see all models struggling with the length extrapolation of all tasks. Exceptionally, Algebraic, Rotary and Relative still achieve perfect performance in Copy. Algebraic and Relative (but *not* Rotary) still remain competitive in Reverse, and all three diverge in Repeat (where basically the length extrapolation required of the decoder is $\times 4$ of the input).
>
> ---
>
> Hope our response satisfactorily addresses your concerns. Otherwise, we'd be happy to engage further.

---

> > ### Comment · Reviewer_PGKE · 2024-08-08
> > **Response to Rebuttal**
> >
> > Dear Authors,
> >
> > Thank you very much for your rebuttal. I have read the Python code and checked the rebuttal response. Currently, I have decided to increase my score from 5 to 6 and my confidence from 2 to 3.
> >
> > However, I still have two additional concerns:
> > * According to the Python Code, the proposed method utilizes learnable parameters, while the baseline RoPE does not. Therefore, my concern is whether the comparison is fair or not.
> > * For the file eval/models/image/cct.py Line 53, the author has atn_fn = self.positional_encoder.adjust_attention((x_maps, y_maps), (x_maps, y_maps), (0.98 ** dists, True)). How do you determine the hyperparameter 0.98 and what does it mean?
> >
> > Overall, the authors have addressed most of my concerns, and I sincerely hope that the author can further explain the above two questions.

---

> > > ### Author Response · Authors · 2024-08-08
> > >
> > > Thanks for taking the time to look through the code, it means a lot.
> > >
> > > * Yes, this is correct! This is also a remark made by reviewer F3X8. Rewording our response to them: the comparison is "fair" in the sense that the sequential version of Algebraic PE (APE) is *exactly* a trainable version of *RoPE*. The comparison is also "fair" in the sense that *RoPE* parameters are indeed *fixed*, but not *randomly fixed* -- they are the product of offline optimization. We're waiting to hear from F3X8 on what experiments could be used to alleviate the effect of these extra trainable parameters -- feel free to share any suggestions or insights.
> > > * The hyperparameter is what we denote as $c$ in Appendix A.1 -- it inserts a minor exponential decay onto the PE matrix to introduce a "locality bias", *i.e.* a progressively smaller weighting of distant positions.

---

> > > > ### Comment · Reviewer_PGKE · 2024-08-08
> > > > **Response to Author**
> > > >
> > > > Dear Authors,
> > > >
> > > > Thank you very much for your quick update.
> > > >
> > > > If there is any update/experiment for the trainable RoPE/non-trainable Algebraic PE, please let me know. I will adjust the score or the confidence according to the experiment setting and results.

---

> ### Author Response · Authors · 2024-08-09
>
> Following your and reviewer F3X8's suggestions, we ran a series of additional experiments where we seek to account for (i) trainability and (ii) initialization. We find that, on average,
> * A frozen APE outperforms a frozen RoPE.
> * A frozen APE outperforms a trained APE.
> * A trainable RoPE outperforms a frozen APE.
> * A trainable APE mimicking the initialization strategy of RoPE outperforms everything else.
>
> Here's the new and old results over the sequential copied verbatim from our response to F3X8. `APE`: Algebraic (seq), `init`: RoPE-like initialization, `train`: trainable. `[b]`: breadth-first, `[d]`: depth-first.
>
>
> | **Task** | APE[init,train] | RoPE[train] | APE | APE[train] | RoPE |
> | ----------- | ------------------ | --------------- | ------------- | --------| -----------------|
> | Copy  | **1** | **1** | **1** | **1** | **1** |
> | Reverse  | **1** | **1** | **1** | **1** | **1** |
> | Repeat  | **1** | **1** | **1** | **1** | **1** |
> | Tree-Copy [b] | **1** | **1** | 2.2 | **1** | 1.9 |
> | Tree-Copy[d] | **1.3** | 1.8 | 2.2 | 2.4 | 3.2 |
> | Rotate[b] | **2.9** | 4.2 | 3.4 |5.2 | 4.9 |
> | Rotate[d] | **2.3** | 2.5 | 3.4 |5.7 | 6.6 |
> | C3[b] | **1.1** | 1.2 | 1.2 |1.5 | 2.0 |
> | C3[d] | **1.9** | 1.9 | 2.1 |2.3 | 2.4 |
> | OP[b] | 2.3 | 2.5 | 2.6 | **1.8** | 2.6 |
> | OP[d] | 20.0 | 21.3 | 19.9 | 29.3 | 41.2 |
>
> So to summarize: APE[init, train] > RoPE[train] > APE > APE[train] > RoPE
>
> In any case, many thanks for encouraging us to do these additional experiments. These empirical findings have helped us consolidate the effect of training and initialization for both RoPE and APE, as well as further reinforce the empirical strengths of Algebraic encodings.

---

> > ### Comment · Reviewer_PGKE · 2024-08-11
> > **Response to Author**
> >
> > Dear Authors,
> >
> > Thank you very much for your reply. After reading the updated experiments, I have more confidence to this work. Therefore, I increase the confidence to 4

---

### Official Review · Reviewer_18aa · 2024-07-10

**Soundness:** 4
**Presentation:** 2
**Contribution:** 4
**Rating:** 8
**Confidence:** 4

**Summary:**

Current techniques for specifying relative position encodings are often ad-hoc or clearly fit to specific problems. This paper attempts to resolve this by creating theoretically grounded positional encodings via describing the underlying algebraic structure of the positions in the input as a group and then finding some subgroup of the group of invertible matrices that is isomorphic to the position group. This results in more effective positional encodings than current approaches.

**Strengths:**

Reconstructing the existing approach towards relative positional encodings for sequences is a strong sign of the effectiveness of this approach.

Evaluation is done fairly with “same budgets (both memory and time).” The empirical results are strong.

**Weaknesses:**

Major clarity issue: while the concepts of a group and group homomorphism are both well explained, the specific reasoning behind wanting to multiply elements in an dot product attention with elements of a matrix group corresponding to the algebraic structure of the underlying data remains highly unmotivated. More explanation provided for this would substantially help the clarity and motivation of the paper. I believe lines 76-79 are attempting an explanation along these lines, but the reference was too oblique to be easily understood by readers who have not yet read the rest of the paper (at least it was to me).

120: “can asymptotically be obtained using O(⌈log2(p)⌉) matrix steps” requires a clear explanation that this is about computing a single embedding

Minor issues
Clarity: On line 49, it might be easier to understand if you write A(p_x X, p_y Y) has the same evaluation as p_x A(X, Y). I was a bit confused at first when reading it.

**Questions:**

Is there any connection between these encodings and the sinusoidal encodings often used for absolute positional encoding?

**Limitations:**

Limitations are adequately explained.

---

> ### Author Rebuttal · Authors · 2024-08-04
>
> Greetings, and thank you very much for your review.
>
> ---
>
> We appreciate your feedback and are glad you saw merit in our approach. We will try to address your concerns below.
>
> >  `wanting to multiply elements ... remains highly unmotivated`
>
> We hear your criticism. The multiplicative approach is the de facto standard ever since RoPE. We see our paper as basically (i) a necessary theoretical justification of why this really is appropriate, and (ii) a guideline for how to go from here. Here's a brief and condensed explanation:
> 1. A positional encoding $\mathrm{PE}$ is a function $\mathrm{PE} : \mathrm{position} \to \mathrm{content} \to \mathrm{vector}$. In words, given a position (in the ambient space) and a content-vector, it returns a position-and-content-vector. The additive approach suffers from a key conceptual limitation: it is commutative (there's no separation between content and position). That is, content and position are mixed as if being "on an equal basis". The multiplicative approach is making the functional dependency clearer. Given a position, you have a linear map, *i.e.,* a function, from content vectors to content-and-position vectors. As you smoothly change position on the ambient space, so does the *function*. The change is such that the effective difference between two positions (and their functions) depends not on what the positions are, but what the relation between them is (where "relation" is more than just "distance").
> 2. Matrices and linear algebra are a folklore target for groups in representation theory. Upon noticing that the structures we are mostly working with are actually just groups in disguise, matrices make for a very natural modeling choice -- you can't preserve these really meaningful properties with vectors. But now you're left with a positional representation that's a matrix, and a content representation that's a vector. And since the rest of the model works on vectors, you can't but end up with a positions-as-functions interpretation.
>
> The ideas above are extensively detailed in Section 3. So, to put things in context, we sort of start from the fact that multiplicative positional encodings *work*, and we seek to provide insights on *why they do*. Because of that, we didn't think it necessary to explicitly motivate them against additive ones.
>
>
> > ` 120: “can asymptotically be obtained using O(⌈log2(p)⌉) matrix steps” requires a clear explanation that this is about computing a single embedding`
>
> This is incorrect -- it takes $O(\lceil \mathrm{log}_2(p) \rceil)$ steps to obtain **all** embeddings; let us explain how. Suppose you have a matrix $W$ of size ($n$, $d$) where $W_i$ contains the representation of the $i$-th position. The matrix $W' := W_nW^\top$ is of size ($n$, $d$), where $W'_j$ contains the representation of position $n+j$. Concatenate $W$ with $W'$ to obtain a matrix of size ($2n$, $d$) containing the representations of positions 1 to 2n, and repeat the same process. By induction, after k vector-matrix multiplications you will have a matrix of size ($2^k$, $d$).
>
> ---
>
>
> Please let us know if that helps. Many thanks, again.

---

> > ### Comment · Reviewer_18aa · 2024-08-09
> >
> > > Matrices and linear algebra are a folklore target for groups in representation theory. Upon noticing that the structures we are mostly working with are actually just groups in disguise, matrices make for a very natural modeling choice -- you can't preserve these really meaningful properties with vectors. But now you're left with a positional representation that's a matrix, and a content representation that's a vector. And since the rest of the model works on vectors, you can't but end up with a positions-as-functions interpretation.
> >
> > I agree you describe the consequences of this in Section 3, but I do think spelling this out would be useful for motivation.
> >
> > > [discussion regarding line 120]
> >
> > I think a footnote explaining that these matrix multiplications are on exponentially sized matrices would be helpful, or perhaps providing an asymptotic flop usage would be helpful. Obviously GPUs are good at parallelism, but that has limits.

---

> ### Author Response · Authors · 2024-08-09
>
> Both are fair points. We will include the appropriate remarks as suggested.

---

### Official Review · Reviewer_F3X8 · 2024-07-11

**Soundness:** 4
**Presentation:** 4
**Contribution:** 3
**Rating:** 7
**Confidence:** 3

**Summary:**

This study proposes a generalized positional encoding for sequential, tree, and grid-structured data. The high-level view of encoding is algebraically framed, while the implementation of encodings remains simple; roughly, positions are embedded in linear subspaces spanned by the column space of orthogonal matrices. Numerical experiments show its empirical superiority over baseline methods on diverse tasks.

**Strengths:**

- This study presents a clear, high-level (algebraic) view of positional encoding, covering various settings.
- The implementation of the proposed encodings is simple and, thus, accessible to a wide range of practitioners.
- The empirical performances of the proposed encodings are better than baseline methods on most tasks. The gain is very evident for several tasks.

**Weaknesses:**

**Major comments**

The main weakness of this work is that it does not offer why the proposed method works better than the other baseline methods. Section 3.2 presents the equivalence of RoPE and the proposed (sequential) encoding, but the experimental results given in Table 2 are not equal. Particularly, (b) shows there is a noticeable performance gap between Algebraic (seq) and Rotary. The discussion in Section 4.4 is too simple. One possible reason is the difference in the distribution of $\theta_i$. In RoPE, the angles are selected from low to high accordingly, while those of a random orthogonal matrix may be different (probably concentrating around some value).

I suggest that the Authors remove Table 1 and instead include more elaboration on the potential difference between RoPE and the proposed encoding and experimental results.

**Minor comments**
- [line 117] Unfamiliar notation $W^p\ :\ \langle W \rangle$; possibly  $W^p \in \langle W \rangle$?
- [line 120] Unfamiliar notation $[W^0\cdots W^p]$; a little confusing with a matrix product.
- [line 178] Comma missing; $\langle W_1, W_2, \ldots, W_L\rangle$. See also [line 140].

**Questions:**

I'd like the Authors to answer the weaknesses raised above. Plus,
- I'm interested in the case of trainable positional encoding (i.e., position embedding). In image classification tasks, it is also common to use position embedding, which learns a proper encoding over the training process (mentioned in [line 269]?). The proposed algebraic encoding can be readily made trainable, and it'd be interesting to see the impact of trainability on performance.

**Limitations:**

Limitations are adequately presented. For the potential improvements, see Weakness and Questions.

---

> ### Author Rebuttal · Authors · 2024-08-04
>
> Greetings, and many thanks for an insightful review.
>
> ---
>
> We are pleased that you appreciated the clarity of the presentation and the simplicity of the implementation.
>
> Your questions and remarks raise valid points, but we believe there may be a few minor but crucial misunderstandings. We will address these below.
>
> > `... why the proposed method works better than other baseline methods`
>
> Our method parameterizes the entire ambient space (e.g., an infinite sequence, grid, or tree) using a few primitives and their compositions. What this means in practice is that we have a smooth way to obtain functional representations that **consistently** relate positions to one another. This is arguably better than ad hoc encoding schemes -- not just for interpretability or epistemic justification, but also (as the resuts suggest) for concrete performance, at least in the settings investigated. Put simply, we think it works better because it should work better.
>
> > `...equivalence of RoPE and the proposed (sequential) encoding, but the experimental results given in Table 2 are not equal..`
>
> This is a very valid point, and one raised also by reviewer 8qDc, albeit in a different context. There are two different things to consider:
> 1. Algebraic positional encodings *theoretically* encompass Rotary ones. We explain why and when in Section 3.2.
> 2. That said, the two approaches are *implemented* differently, and are subject to different training dynamics. RoPE uses a fixed distribution of angles (as you correctly point out). Why "*these* angles specifically" is subject to interpretation; we hypothesize that the decision is the product of implicit task- and dataset-specific tuning (see also our response to reviewer q8Dc). This might also explain the minimal but evident disparity in performance across tasks. Our system parameterizes the rotation angles in the orthogonal primitives, and allows them to change during the course of training -- this might indeed have a significant effect on the training process (and therefore final performance). A minor remark here is that we do not opt for `random` angles, but rather initialize our primitives to be close to Identity (see Appendix A.1).
>
> > `The discussion in Section 4.4 is too simple.`
>
> We agree. We plan to extend the discussion with relevant remarks and hypotheses if/when given extra space.
>
> > `...trainable positional encoding`
>
> There seems to be a bit of confusion here and the blame is likely on us. The encodings we use **are** actually trainable (see lines 26, 153, 295, Appendix A.1, *i.a.*). We will make this more explicit in a future version.
>
> > `..unfamiliar notation`
>
> We borrow the notation `x : A` from logic/functional programming to denote that `x` is of type `A`, or a member of set `A` if you will. We will add the missing commas in the set builder notation, thank you.
>
> ---
>
> Please let us know whether this addresses your concerns. Thanks again!

---

> ### Comment · Reviewer_F3X8 · 2024-08-08
> **Response to Rebuttal**
>
> Thank you for the rebuttal.
>
> **Insights**
>
> I understand the proposed method has a degree of freedom compared to the ad-hoc ones, and the experiments show better performance empirically. My questions were about the (theoretical and empirical) insights of the proposed positional encoding.
>
> Having more degrees of freedom does not always mean better. That's why ad-hoc positional encodings have been used rather than position embedding, where position vectors are randomly initialized and trained over training. Let's focus on the sequential data, not trees or grids. The proposed encoding method, in this case, is conceptually close to the RoPE with a little more degree of freedom.
>
> > we have a smooth way to obtain functional representations that **consistently** relate positions to one another.
>
> Does it also claim that the proposed method is consistent, but RoPE is not? I assume not (i.e., both are consistent). The better empirical results presented in the experiments indicate that the proposed encoding acquires a better position representation. I'd like the authors to provide any insights obtained from the comparison between the RoPE and the position representation acquired by the proposed method. This also relates to my mention of the distribution of $\theta$. The impact on the training process, as the authors mentioned, is also an interesting point.
>
> **Trainability**
>
> If the proposed encoding is tested with trainable vectors, then it seems to be fairer to make other position encoding methods trainable in a straightforward way. In addition, from section 3, the proposed method should work without trainability (and trainability makes it better). The contributions of trainability should also be of interest to readers.
>
> **Notation**
>
> Personally, I don't think that logic/functional programming notation is very common among most people interested in positional encoding, from researchers to practitioners. I suggest the authors define it or replace it with reader-familiar notations to maximize the impact of the paper.

---

> > ### Author Response · Authors · 2024-08-08
> >
> > > `Does it also claim that the proposed method is consistent, but RoPE is not?`
> >
> > No. Both do the same thing. The difference is that one is tied to sequences, and presented in a "it's good because it works, it works because it does" manner, whereas the other is general, theoretically motivated and reached naturally through group representation theory. Let us clarify that we do not seek to "compete" with RoPE -- we seek to formalize and extend it.
> >
> > > `... fairer to make other position encoding methods trainable in a straightforward way ...`
> >
> > This is very tricky.
> >
> > Theoretically:
> > 1. Algebraic PE (APE) over sequences is **exactly** that: RoPE over sequences made trainable in a straightforward and theoretically valid way.
> > 2. It feels strange that our methodology being trainable obliges us to extend a historical non-trainable alternative for the sake of "fairness".
> >
> > Practically:
> > 1. RoPE is awkward to make trainable since it is parameterized over *rotation angles*. Gradient-based optimization over periodic parameters is not trivial.
> > 2. Algebraic is better-behaved (for training purposes); it is parameterized over (orthogonal) matrices. Angles and periodicity become implicit. Note: this assumes and requires the "small initialization" we describe in Appendix A.1.
> >
> > That said, we're not trying to evade your point, because it really is a good one. How would you suggest we proceed to produce the empirical insights you have requested?
> >
> > We could try out a trainable RoPE, *.e.g.* following the implementation of popular online sources. We could also try out a non-trainable, randomly initialized APE (but note that this wouldn't be "fair" either; RoPE angles are *fixed*, but that doesn't mean they haven't been experimentally *optimized*). Would these experiments help

---

> > > ### Comment · Reviewer_F3X8 · 2024-08-08
> > >
> > > Thank you for your response. I'd like first to clarify that I appreciate the unified framework given by algebraic encoding.
> > >
> > > **Insights.**
> > > Once things become trainable and perform better than ad-hoc ones, we become interested in what was achieved by the training (what was lacking in the ad-hoc ones). That's why I'm asking about any insight obtained from the algebraic encoding compared to RoPE.
> > >
> > > **Equivalence**
> > > My understanding about the algebraic encoding for sequential data is that it is conceptually equivalent to RoPE but different in that they are implemented by an orthogonal matrix $W$, or from the spectral viewpoint a) it has a (trainable) orthogonal matrix $Q$ and b) trainable angles. If a) has no effect, then yes, the algebraic encoding is a trainable version of RoPE.
> > >
> > > Let's assume a) has no effect (Yes, this is reasonable as $Q$ is invertible, and other trainable matrices can cancel this effect.)
> > >
> > > > RoPE is awkward to make trainable since it is parameterized over rotation angles. Gradient-based optimization over periodic parameters is not trivial.
> > >
> > > Practically, the implementation is trivial, as the authors may understand; using PyTorch, we just need a trainable tensor nn.Parameters([$\theta_1$, $\theta_2$, ..., $\theta_s$]). Theoretically, the optimization of periodic parameters may be non-trivial, but why is the optimization of algebraic encoding empirically successful? Assuming a) has no effect, optimization of algebraic encoding is an optimization of periodic parameters. The authors' answer "2) ... angles and periodicity become implicit" is personally very interesting. Will you provide any more comments about this?
> > >
> > > > It feels strange that our methodology being trainable obliges us to extend a historical non-trainable alternative for the sake of "fairness".
> > >
> > > I understand your claim, but I consider this beneficial as it allows the reviewers to confirm that the proposed method is certainly impactful. Generally speaking, if the proposed method performs worse than a trivial extension of a baseline method, the impact of the proposed method is less than the trivial extension (from the performance perspective).
> > >
> > > I understand that the authors claim that the proposed method is equivalent to a trainable RoPE and don't claim beyond it. So it is OK (i.e., In a very harsh view, the proposed method is a trivial extension of RoPE for sequential cases). Of course, the essence of algebraic encoding is its generality; it encompasses RoPE, provides a unified view covering sequential, tree, and grid structures, and has consistently good empirical performance.
> > >
> > > I appreciate this work and thus keep my score. I would also appreciate it if the authors could share some feedback regarding my two questions above (i.e., insights/optimization of orthogonal matrices). The questions are out of my curiosity. The authors may prioritize the discussion with other reviewers.

---

> > > > ### Author Response · Authors · 2024-08-08
> > > >
> > > > We hear your points and appreciate the discussion. We'll return in a couple of hours with additional results and a few comments.

---

> ### Author Response · Authors · 2024-08-09
>
> Apologies for the delay, and thank you for your patience.
>
> > `angles and periodicity become implicit`
>
> Intuitively, RoPE optimizes over trigonometric functions, the derivatives of which are also trigonometric functions, which are (i) not monotonic and (ii) periodic. This *should*, in theory at least, pose optimization issues. In contrast, APE optimizes over (parameterized) orthogonal matrices, the derivatives of which should be better behaved.
>
> > `insights`
>
> We have conducted the experiment you requested (*i.e.*, turning RoPE trainable naively and, *vice versa*, freezing APE). To our dismay, we found that:
> * RoPE diverged only twice across experiments.
> * In most cases, a frozen APE outperformed a trained APE.
> * In most cases, a trained RoPE outperformed a frozen APE.
>
> To provide an empirical correspondence between RoPE and APE, as requested, we then devised a back-and-forth algorithm between the two in the sequential case. We detail the process below.
>
> ### **Parameterizing APE**
> First, our parameterization of orthogonal primitives follows the standard matrix exponentiation trick on skew symmetric matrices:
> 1. Start from an upper triangular square matrix $A$. (your trainable parameter)
> 2. Obtain a skew symmetric $B := A - A^\top$.
> 3. Take the matrix exponent $C := e^{B}$. $C$ is orthogonal.
>
> ### **RoPE to APE**
> To go from RoPE to APE:
> 1. Start from a sequence of RoPE angles $\Theta := [\theta_1, \theta_2, \theta_3, \dots, \theta_n]$.
> 2. Block-diagonalize $\Theta$ into
> $$
> C := \begin{bmatrix}
> cos\theta_1 & -sin\theta_1 & 0 & 0 & \dots \\\\
> sin\theta_1 & cos\theta_1 & 0 & 0 & \dots \\\\
> 0 & 0 & cos\theta_2 & -sin\theta_2 & \dots \\\\
> 0 & 0 & sin\theta_2 & cos\theta_2 & \dots \\\\
> \vdots & \vdots & \vdots & \vdots & \ddots
> \end{bmatrix}
> $$
> 3. Stop here if not interested in parameterization. Otherwise, obtain the matrix logarithm of $C$, $ B:= \mathrm{log}(C)$ using a numerical solver.
> 4. Find $A$ such that $\mathrm{MSE}(B, A - A^\top)$ is minimized. $B$ is not positive-definite so it does not admit a Cholesky decomposition -- this means you have to use an iterative approximation. We find it's easy to obtain one with minimal error within seconds using gradient-based optimization.
>
> ### **APE to RoPE**
> To go from APE to RoPE (this is a practical variant of what's described in Section 3.2)
> 1. Start from orthogonal matrix $C$.
> 2. Perform the polar decomposition $C = UP$. Since $C$ is invertible, $U$ is unique, and denotes a rotation/reflection along the axes specified by $P$. For $P \equiv 1$, $U$ is block-diagonal and corresponds to $C$ above (*i.e.* the absolute angles of its eigenvalues correspond to $\Theta = [\theta_1, \theta_2, \dots]$).
>
>
> ### **Results**
>
> Besides explicating the relation between APE and RoPE, this back-and-forth allows us to do something more: we can now *initialize* APE with a parameterization that produces the angles used by RoPE! Doing so allows us to examine the effects of trainability *in juxtaposition with initialization*. We find that in almost every setup a trainable APE initialized as RoPE **outperforms all other models**. Together, these results empirically confirm both **your claim** that RoPE can do better if trained, and **our claims** that (i) gains are also due to initialization, and (ii) RoPE has been "tuned" offline.
>
> For the sake of comparison, we summarize all results in the table below. `APE`: Algebraic (seq), `init`: RoPE-like initialization, `train`: trainable. `[b]`: breadth-first, `[d]`: depth-first. We do not include the 2 diverging RoPE runs so as not to obfuscate the results.
>
> | **Task** | APE[init,train] | RoPE[train] | APE | APE[train] | RoPE |
> | ----------- | ------------------ | --------------- | ------------- | --------| -----------------|
> | Copy  | **1** | **1** | **1** | **1** | **1** |
> | Reverse  | **1** | **1** | **1** | **1** | **1** |
> | Repeat  | **1** | **1** | **1** | **1** | **1** |
> | Tree-Copy [b] | **1** | **1** | 2.2 | **1** | 1.9 |
> | Tree-Copy[d] | **1.3** | 1.8 | 2.2 | 2.4 | 3.2 |
> | Rotate[b] | **2.9** | 4.2 | 3.4 |5.2 | 4.9 |
> | Rotate[d] | **2.3** | 2.5 | 3.4 |5.7 | 6.6 |
> | C3[b] | **1.1** | 1.2 | 1.2 |1.5 | 2.0 |
> | C3[d] | **1.9** | 1.9 | 2.1 |2.3 | 2.4 |
> | OP[b] | 2.3 | 2.5 | 2.6 | **1.8** | 2.6 |
> | OP[d] | 20.0 | 21.3 | 19.9 | 29.3 | 41.2 |
>
> So to summarize: APE[init, train] > RoPE[train] > APE > APE[train] > RoPE
>
> ---
>
> To us this seems like a very interesting find. Does this help answer your questions?
>
> In any case, many thanks for encouraging us to do these additional experiments. These empirical findings have helped us consolidate the effect of training and initialization for both RoPE and APE, as well as further reinforce the empirical strengths of Algebraic encodings.

---

> > ### Comment · Reviewer_F3X8 · 2024-08-11
> >
> > Thank you for providing a new experiment and analysis.
> >
> > So, we now have a clearer view of the APE, particularly its connection to RoPE and the importance of the initiative. I'm looking forward to reading the paper completed by integrating this discussion. I increase my score by one.

---

### Official Review · Reviewer_8qDc · 2024-07-12

**Soundness:** 4
**Presentation:** 4
**Contribution:** 4
**Rating:** 8
**Confidence:** 4

**Summary:**

This paper provides a group-theoretic framework to generate positional encodings which reflect and preserve the underlying algebraic structure of the data by instantiating homomorphisms between a syntactic denotation of the data structure and a semantic interpretation using orthogonal matrices.  The authors demonstrate that their methodology is able to encode a variety of structures, including sequential data,  grids, and trees.  They demonstrate that in the case of sequential data, they recover the rotary positional encoding scheme.  The authors provide a thorough empirical evaluation of their framework against multiple baselines, demonstrating the efficacy.

**Strengths:**

- The paper is excellently written and provides many fascinating group-theoretic insights into how to effectively encode structured data for use in a language model.
- The work is original in that it provides a universal group-theoretic framework for generating encoding schemes for various positional encodings that reflect various structural forms of the data.  It demonstrates how their framework subsumes some existing works, such as the rotary encoding scheme of Su et al.
- I believe the framework is a very important contribution to the sound and systematic construction of encoding schemes that preserve the structure of various data types, and that the framework facilitates lots of interesting future work.

**Weaknesses:**

- I believe it would benefit the paper to provide a few explanations of the empirical results regarding when other encoding schemes outperform the algebraic encoding or achieve the same scores (e.g. why might this be the case that rotary outperforms algebraic in row 1 of Table 2a) and why might relative, rotary, and algebraic have the same performance on the copy task in Table 2a) ?).

**Questions:**

I have no specific questions.

**Limitations:**

The authors have sufficiently addressed the limitations of the work.

---

> ### Author Rebuttal · Authors · 2024-08-04
>
> Thank you so much for your feedback.
>
> ---
>
> We are thrilled that you found our work insightful and original, and that you consider it an important contribution to the field; this means a lot.
>
>
> You’re absolutely right that we did not provide an in-depth explanation of the findings in Table 2. We plan to address this in a future iteration or the camera-ready version. For clarity, here are the main takeaways:
>
> 1. The (sequence) copy task is somewhat trivial—it mainly tests if the PE system learns position-based addressing without relying on content. This is essentially a sanity check, and all systems pass it except for Absolute (Sinusoidal performs only marginally worse than perfect). We hypothesize that Absolute lags behind due to overfitting: it assigns a unique representation to each position, which over-parameterizes the system without offering any real inductive biases.
> 2. This hypothesis is supported by the tree copy experiment. When teaching a transformer to copy-paste, the structure of the content becomes a confound. Because structure is a confound, point-wise ad hoc position representations are likely to lead to overfitting. Indeed, Absolute fails to provide a sufficiently general position representation strategy and achieves the worst performance once more.
> 3.  We highlight in red the numbers corresponding to the best-in-class setups and underline those within a 95% confidence margin of the best-in-class. Apart from random variations, we consider red and underlined setups as equally effective. Notably, Rotary and Algebraic are closely aligned in the sequential case (as expected). The slight advantage of Rotary in WMT and the preference for Algebraic elsewhere might be due to an implicit **selection bias** in the hyper-parameters of Rotary. We suspect that RoPE authors conducted iterative experiments, which could make it implicitly tuned for this specific task and dataset. In contrast, our experiments are not fine-tuned with a specific benchmark in mind; our contribution is first and foremost theoretical, and the experiments serve as a first empirical proof-of-concept meant to showcase the methodology's strength rather than compete in a *specific* benchmark.
>
> ---
>
> We hope this clarifies your question. Thank you once again for your review and engagement.

---

> > ### Comment · Reviewer_8qDc · 2024-08-09
> >
> > I thank the authors for their response; it resolves my concern.  I look forward to future work in this area.

---

### Official Review · Reviewer_xNaW · 2024-07-12

**Soundness:** 4
**Presentation:** 3
**Contribution:** 4
**Rating:** 7
**Confidence:** 3

**Summary:**

This paper presents a framework to derive positional encodings based on algebraic specifications for various structures such as sequences, grids, trees, and their compositions. The approach is evaluated on multiple training tasks involving sequence, image, and tree inputs, demonstrating performance that is comparable to or better than existing positional encodings.

**Strengths:**

- Unlike many papers that focus on a single type of positional encoding for a specific data structure, this paper offers a unified framework applicable to multiple structures. This is a significant contribution as it provides a versatile tool for handling diverse data types.

- The experimental results are promising, particularly with algebraic (tree) and algebraic (grid) encodings significantly outperforming existing methods.

- The paper also discusses the applicability of the framework to other structures beyond sequences, grids, and trees. This is definitely very promising and looking forward to future directions in this space.

**Weaknesses:**

One thing to further improve the paper is to have some experiments with compositional structures. For example, the authors can consider some table based tasks that combine grid with time series data.

**Questions:**

Is there any impact on the inference times? It is not clear from the description how computationally hard it is to compute the different algebraic encodings and the size of the encodings compared to the existing encodings.

How does the complexity of the encoding scale with the size of the input?

**Limitations:**

Yes, the authors discuss the limitations of this work.

---

> ### Author Rebuttal · Authors · 2024-08-05
>
> Many thanks for your thoughtful and constructive review.
>
> ---
>
> We are very glad you found our framework interesting and the results promising, and that you appreciated the potential for future directions.
>
> We acknowledge your remark about (more complex) compositional structures. This is something we have discussed also internally, but ended up refraining from, for one main reason: the comparative evaluation would require us to make some subjective design choices on the "competing" frameworks (because they are not meant to handle such structures). This would give room for a critical counter-argument along the lines of "why did you try this and not that", which could easily derail the discussion into an exhaustive evaluation of *how* and *how not* to use foreign works properly.
>
> To answer your question regarding inference times: assuming that, after training, positional encodings are precomputed and stored in a buffer (as they should), the computational cost is just an extra two matrix-vector multiplications per transformer layer: between queries/keys and their respective encodings.
>
> There's two gotchas unique to the tree case:
> 1. The number of unique positions grows very fast for dense and large trees, which might necessitate dynamically computing representations *given* a tree. Practical workarounds would be to store most common/shallow positions and only compute rare/deep ones as needed.
> 2. If the tree is sparse (i.e., not all of its positions are filled), building up the matrix of positional encodings will require an additional indexing step that pieces together all of the occupied positions (think like the forward pass of an embedding layer).
>
> For the sake of clarity and transparency, we report **inference times** for the test set of WMT14 under the same settings and hardware for all sequential encoding models below (average of 3 repetitions):
>
> | Sinusoidal | Absolute | Relative | Rotary | Algebraic |
> |------------|----------|----------|--------|---------|
> |     105.1     |    108.1 | 112.9     |   127.9     |   112.7   |
>
> Evidently, inference time is only marginally affected, and is in fact better than RoPE; the latter uses a vector-vector formulation that is optimized for memory efficiency, but requires additional slicing operations. Inference in the remaining experiments is not auto-regressive so there's no insights on asymptotic behavior to gain from them.
>
> When it comes to **memory complexity**, and assuming all encodings are precomputed (rather than dynamically generated), the parameter count scales quadratically with attention dim, and linearly with number of *unique* positions in the ambient structure. Additionally, there's a linear dependency on number of attention heads and transformer layers, assuming no weight sharing. In our experimental setup, we share positional encodings across transformer layers but not across attention heads.
>
> ---
>
> Thanks again, and we hope this answers your questions.

---

### Decision · Program_Chairs · 2024-09-25

**Decision:**

Accept (spotlight)

**Comment:**

At this point transformers have been applied to a wide variety of data, which frequently have some underlying structure.  E.g. 1D (sequence), 2D (image), or 3D (video) cartesian spaces, as well as syntax trees (code).  This paper addresses an important gap in the literature, namely how practitioners should systematically design position encodings to handle these spaces, or new spaces that might arise.  The authors base their analysis in group theory, and show how position encodings for a variety of spaces can be constructed from orthogonal matrices.  As a bonus, the widely used ROPE encodings can be realized as a special case of the more general construction.

All of the reviewers agreed that this was important theoretical work with practical applications, and was well-presented.  Several reviewers commented that the experimental evidence was not as strong as the theory.  I concur with this opinion, but the experiments are strong enough to make this paper a clear "accept".  I hope the authors make a stronger experimental case in future work.  The technique is also somewhat difficult and/or costly to implement for trees, which IMO is the biggest weakness.

I am recommending this paper for a spotlight because I think it will be of immediate practical interest to a wide variety of NeurIPS attendees.  I am tempted to recommend it for an oral, but I don't think the experimental evidence quite justifies that.